# Isolation of Anti-Inflammatory and Epithelium Reinforcing *Bacteroides* and *Parabacteroides* Spp. from A Healthy Fecal Donor

**DOI:** 10.3390/nu12040935

**Published:** 2020-03-27

**Authors:** Kaisa Hiippala, Veera Kainulainen, Maiju Suutarinen, Tuomas Heini, Jolene R. Bowers, Daniel Jasso-Selles, Darrin Lemmer, Michael Valentine, Riley Barnes, David M. Engelthaler, Reetta Satokari

**Affiliations:** 1Human Microbiome Research Program, Faculty of Medicine, University of Helsinki, 00290 Helsinki, Finland; veera.kainulainen@helsinki.fi (V.K.); maiju.suutarinen@helsinki.fi (M.S.); tuomas.heini@helsinki.fi (T.H.); 2Translational Genomics Research Institute, Pathogen and Microbiome Division, Flagstaff, AZ 86001, USA; jbowers@tgen.org (J.R.B.); djasso-selles@tgen.org (D.J.-S.); dlemmer@tgen.org (D.L.); mvalentine@tgen.org (M.V.); rbarnes@tgen.org (R.B.); dengelthaler@tgen.org (D.M.E.)

**Keywords:** bacteroides, gut homeostasis, host-microbe interactions, immunomodulation, LPS, next-generation probiotics

## Abstract

Altered intestinal microbiota is associated with systemic and intestinal diseases, such as inflammatory bowel disease (IBD). Dysbiotic microbiota with enhanced proinflammatory capacity is characterized by depletion of anaerobic commensals, increased proportion of facultatively anaerobic bacteria, as well as reduced diversity and stability. In this study, we developed a high-throughput in vitro screening assay to isolate intestinal commensal bacteria with anti-inflammatory capacity from a healthy fecal microbiota transplantation donor. Freshly isolated gut bacteria were screened for their capacity to attenuate *Escherichia coli* lipopolysaccharide (LPS)-induced interleukin 8 (IL-8) release from HT-29 cells. The screen yielded a number of *Bacteroides* and *Parabacteroides* isolates, which were identified as *P. distasonis*, *B. caccae*, *B. intestinalis*, *B. uniformis*, *B. fragilis*, *B. vulgatus* and *B. ovatus* using whole genome sequencing. We observed that a cell-cell contact with the epithelium was not necessary to alleviate in vitro inflammation as spent culture media from the isolates were also effective and the anti-inflammatory action did not correlate with the enterocyte adherence capacity of the isolates. The anti-inflammatory isolates also exerted enterocyte monolayer reinforcing action and lacked essential genes to synthetize hexa-acylated, proinflammatory lipid A, part of LPS. Yet, the anti-inflammatory effector molecules remain to be identified. The *Bacteroides* strains isolated and characterized in this study have potential to be used as so-called next-generation probiotics.

## 1. Introduction

The human intestine harbors a complex bacterial ecosystem of which 90% of the species belong to the phyla Firmicutes and Bacteroidetes [1,2,3]. Most individuals carry approximately 200 different bacterial species with varying abundance in their gastrointestinal tract but only around 30% of the species, the so-called common core microbiota, are shared between subjects [4,5]. Together with hosts’ genetic and immunological background, environmental factors such as diet, use of antibiotics and other medication and hygienic conditions affect the gut microbiota composition [6]. Dysbiotic, unbalanced intestinal microbiota composition with enhanced proinflammatory capacity is characterized by reduced species richness and diversity as well as reduced microbial stability [7]. Alterations in the abundance of health-associated, short-chain fatty acid-producing Firmicutes and proinflammatory, lipopolysaccharide-(LPS-)containing pathobionts, such as gamma-Proteobacteria, have been linked to dysbiosis [8]. Moreover, loss of intestinal immune homeostasis and barrier function is associated with systemic and intestinal diseases, such as inflammatory bowel disease (IBD) and irritable bowel syndrome (IBS) [9]. Several studies have shown that IBD patients have decreased abundance of mucosal or fecal *Bacteroides* in comparison to healthy subjects [10,11,12].

Manipulation of the gut microbiota towards a more balanced and optimal microbial community to promote human health has received increasing interest in recent years. In addition to traditional probiotics and prebiotics [13], new therapeutic approaches ranging from total microbiota manipulation by fecal microbiota transplantation (FMT, [14]) to narrow-spectrum microbiota manipulation by targeted inhibition of metabolic pathways in a specific bacterial group [15] are being investigated. The great success of FMT in treating recurrent *Clostridioides difficile* infection (rCDI) and promising findings for several other conditions [14] together with studies on microbiota dysbiosis have paved the way for so-called next generation probiotics and the development of bacteriotherapy [8]. In IBD, where the treatment success of FMT has been variable [16,17], using defined bacterial communities as a potent bacteriotherapy could provide more consistent efficacy compared to FMT which has significant variation between donors [18]. Indeed, in a murine model of IBD, treatment with triple *Bacteroides* strain combination, and specifically *B. ovatus*, decreased inflammation and was more effective at reducing colitis than traditional FMT [19]. Recently, the U.S. Food and Drug Administration (FDA) approved an Investigational New Drug (IND) application concerning the safety and efficacy of *B. ovatus*, *B. thetaiotaomicron* and *B. vulgatus* as a treatment for rCDI [19]. In a study about encapsulated delivery of FMT, *Bacteroides* and *Parabacteroides* were among the species that primarily contributed to the donor engraftment, potentially serving as keystone species in maintaining donor-like microbiota [20].

Bacteroidetes is the second largest phylum after Firmicutes in the healthy adult human gut [1,2,3] and its relative proportion of the total microbiota is higher in mucosa than in feces [21]. Genus *Bacteroides*, accounting for approximately 30% of the intestinal microbiota, contains most predominant species within the Bacteroidales order [22]. *Bacteroides* spp. are equipped with different mechanisms in order to adapt to the harsh environment of the intestine, such as metabolizing many diet- and host-derived polysaccharides, oxygen toleration using cytochrome bd oxidase, and vast expression of cell surface structures [23]. *Bacteroides* species include many important opportunistic pathogens, but as essential members of a balanced microbiota they are considered to be health-maintaining. This is due to their ability to reinforce the epithelial barrier and ameliorate inflammation by producing anti-inflammatory molecules such as polysaccharide A (PSA), sphingolipids and outer membrane vesicles (OMV) for the transport of aforementioned molecules to the epithelium via the thick mucus layer [24,25,26,27]. Thus, *Bacteroides* spp. could be promising bacteriotherapeutic candidates for use in clinical and nutritional applications, but more research on this matter is needed. Safety assessment of strains needs to be thorough because some representatives of the genus are also important opportunistic pathogens.

In this study, we aimed to isolate intestinal commensal bacteria with anti-inflammatory capacity from a healthy FMT donor and set up a high-throughput in vitro screening assay for this purpose. In the screening, we assessed the potential of freshly isolated gut bacteria to attenuate LPS-induced interleukin 8 (IL-8) release from enterocytes i.e., to alleviate inflammatory reaction caused by LPS. The promising isolates from the screen were purified and after a thorough validation of their anti-inflammatory capacity, they were tentatively identified by partial 16S rRNA gene sequencing. Isolates representing *Bacteroides* and *Parabacteroides* spp. were selected for further studies, which included whole genome sequencing and assessment of adhesive and epithelium reinforcing properties.

## 2. Materials and Methods

### 2.1. Bacterial Isolation and Tentative Identification

Feces from a healthy pre-screened adult, who acted as a donor for FMT, were freeze-stored as a saline-10% glycerol solution at −80 °C using a protocol that is successfully used for fecal banking for FMT [28]. The preparation was done under ambient air and the sample was frozen within 1.5 h from defecation. The use of the donor was approved by the Ethics Committee of Hospital District of Helsinki and Uusimaa Finland (DnroHUS124/13/03/01/11). The donor, described in a previous study [21], provided a written informed consent. The donor was a healthy Finnish female, aged 42 years, with a normal body mass index (BMI < 25), no gastrointestinal symptoms and no antibiotic use for the past 6 months. The donor passed all the screening tests for fecal donors [28]. All experiments were carried out in accordance with relevant guidelines and regulations. The frozen fecal solution was thawed in 4–5 hours, serially diluted in phosphate-buffered saline (PBS) and cultivated on Brucella agar (Sigma Aldrich, St. Louis, MO, USA) supplemented with 5% sheep blood (Bio Karjalohja Oy, Karjalohja, Finland), Gifu anaerobic medium (GAM; Nissui Pharmaceutical Co., Ltd., Tokyo, Japan) and reinforced clostridial medium (RCM; Becton Dickinson, Sparks, USA) under anaerobic conditions using an anaerobic chamber containing 85% N_2_, 10% CO_2_, and 5% H_2_ at 37 °C (Concept Plus anaerobic workstation, Ruskinn Technology Ltd., Bridgend, UK). The anaerobic chamber automatically purged gas inside the chamber when needed. Palladium catalyst inside the chamber scavenged any residual O_2_ and anaerobic color indicator strips were used to verify anaerobic conditions. All media were reduced in the anaerobic atmosphere inside the anaerobic workstation for 24 hours before the cultivation. Separate colonies were picked over 72 hours and re-streaked on new agar plates. Each isolate was cultivated in appropriate broth or on agar, i.e., the medium used originally to isolate the bacterium, for the preliminary screening of attenuation capacity. Frozen stocks were prepared from the same bacterial culture and stored at −80 °C. The screening of anti-inflammatory isolates (described in detail in paragraph 2.3) was first carried out in 96-well microtiter plates using one replicate due to the large number of isolates. Next, the screening was repeated for potentially positive isolates using three replicates. The potentially positive isolates were re-streaked on agar plates until pure cultures were obtained. The pure cultures were subjected to Gram-staining and microscopy to evaluate the purity of the cultures. Partial 16S rRNA gene sequencing was used for tentative identification, for which bacterial mass from each isolate was resuspended in TE buffer (10 mM Tris-HCl, 1 mM EDTA, pH 8) and heated to 95 °C for 15 min to break the cells. The resulting cell lysates were used as the template in PCR amplification of the partial 16S rRNA gene using 27F DegL (5′-AGR GTT YGA TYM TGG CTC AG-3′) forward and Pd (5′-GTA TTA CCG CGG CTG CTG-3′) reverse primers. Sanger sequencing of the PCR product using the forward primer was carried out in the Institute of Biotechnology core facility, University of Helsinki. The partial 16S rRNA gene sequences were compared to NCBI 16S ribosomal RNA sequences database using BLASTn to acquire tentative identification at the genus level.

### 2.2. Epithelial Cell Lines

The human colonic epithelial cell lines Caco-2 (ACC 169) and HT-29 (ACC 299) were acquired from German Collection of Microorganisms and Cell Cultures (DSMZ) and grown at 37 °C in an incubator under oxic atmosphere with 5% CO_2_. Cells were passaged after reaching 80% confluence using HyClone HyQTase (GE Healthcare Life Sciences, Marlborough, MA, USA) to detach the cells and passage numbers up to 28 were used in the experiments. Caco-2 cells were cultivated in RPMI 1640 medium (Sigma-Aldrich, USA) supplemented with heat-inactivated (30 min at 56 °C) fetal bovine serum (20%; FBS; Gibco, UK), nonessential amino acids (1%, NEAA; Lonza, Bornem, Belgium), 15 mM HEPES (Lonza, Belgium), 100 U mL^−1^ penicillin and streptomycin (PEST; Lonza, Belgium) and 2 mM L-glutamine (Lonza, Belgium). HT-29 cells were grown in McCoy 5A (Lonza, Belgium) medium containing 10% FBS and 100 U mL^−1^ PEST.

### 2.3. Screening of Anti-Inflammatory Capacity of the Isolates

The capacity of bacterial isolates to attenuate LPS-induced IL-8 production in HT-29 cells was measured as previously described [29], but the assay was adjusted to high-throughput 96-well plate format. HT-29 cells were seeded 10,000 cells per well onto 96-well microplates for the attenuation assay. Broth cultures of bacteria were washed once with McCoy 5A medium supplemented with FBS (10%) and resuspended in the same medium. The bacterial mass of isolates grown on solid agar was collected and resuspended in McCoy 5A medium with FBS supplement. The bacterial suspensions were adjusted to OD_600nm_ 0.25 corresponding approximately to 10^8^ cells/mL and 100 µl of this suspension was added onto 8 days post-plating HT-29 cells and incubated at 37 °C for 1 hour under oxic atmosphere with 5% CO_2_. The bacterial concentration used in the experiments (final concentration of 10^7^ cells/mL) was considered biologically relevant for in vitro assays as the amount of *Bacteroides* spp. in the intestine could theoretically be 5 × 10^9^/mL up to 3 × 10^10^/mL. The abundance of Bacteroidetes in a healthy microbiota has a lot of inter-individual variance, but in general it is up to 30% in feces and the bacterial content in the stool is approximately 10^11^/g of feces [30]. The percentage of *Bacteroides* spp. in the feces of the donor used in this study was approximately 5% [21]. McCoy 5A medium containing 10% FBS was used as the control. Afterwards, the bacterial suspensions were removed from the HT-29 monolayer and 200 µl McCoy 5A medium with 1 ng/mL of *E. coli* LPS (Sigma Aldrich, USA) was added. After a 4-hour incubation with LPS, supernatants were collected and IL-8 levels were measured by using an ELISA assay (BD OptEIATM Set, San Diego, USA). The use of aerobic conditions with increased CO_2_ (5%) in the screening assay using anaerobic bacteria was considered appropriate due to the short activation time i.e., 1-hour incubation on the HT-29 cells before removing the bacterial cells and adding LPS. It is also known that anaerobic bacteria can tolerate oxygen for 1 hour up to 72 hours depending on the species [31], and *Bacteroides* spp. sustain aerobic conditions better compared to other anaerobic gut bacteria [32].

The IL-8 levels were calculated using four parametric logistic curves and the effect of the isolates on the LPS-induced IL-8 production was assessed by comparing the levels to that of the LPS control. In the first screening only one replicate (well) per isolate was used and criteria for positive isolate was a reduction of IL-8 level lower than the LPS control. No statistical testing was applied as only one replicate per isolate was tested. In the second round of screening, three replicates (wells) per isolate were used and the average result of three replicates per sample was compared to the LPS control by using a t-test, and all isolates that showed a significant (*p* < 0.05) reduction in the IL-8-levels as compared to the control were selected for further testing. No secondary statistical analysis by using a post-hoc test against false positives was taken because all isolates would subsequently undergo validation by using biological testing, and therefore loose statistical criteria were applied at the screening stage. Finally, the attenuation assay was repeated with the purified, potentially positive isolates several times. IL-8 assay was only performed using the HT-29 cell line due to defects in the TLR4 signaling of Caco-2 cell line which caused unresponsiveness to LPS stimulation [33,34,35].

The spent medium attenuation assay was carried out similarly as described above. The positive, anti-inflammatory action-exerting isolates were all grown in GAM broth under anaerobic conditions and the OD_600nm_ was adjusted to 1.0. The adjusted bacterial suspensions were centrifuged (10,000 RPM, 3 min.) to pellet the cells and the supernatants were filtered with 0.2 µm filter to remove any remaining bacterial cells. The supernatants were diluted to 1:4 and 1:2 using McCoy 5A medium supplemented with FBS (10%). GAM and McCoy 5A were included in the assay as controls.

### 2.4. Bacterial Adhesion to Cell Lines and Mucus

The adherence of bacterial isolates to Caco-2 and HT-29 cell lines (8 days post-plating) and mucus was studied as described previously [36,37]. Four technical replicates (parallel wells) were used in each experiment. Bacteria were grown in an appropriate medium supplemented with 10 μL mL^−1^ of [6′^-3^H]thymidine (17,6 Ci mmol^−1^, Perkin Elmer, Waltham, USA) to metabolically radiolabel the cells. All isolates within the same species were grown identically and tested for adhesion in the same experiment. For epithelial adherence assay, HT-29 and Caco-2 cells were seeded 10,000 cells per well onto 96-well microplates. To assess the adherence to intestinal mucus (separate assay from the cell lines), porcine mucus (Sigma-Aldrich, 50 μg well-1 in PBS) was allowed to adsorb to Maxisorp microtiter plate wells overnight at 4 °C. [3H]Thymidine-labelled bacterial cells were washed with an appropriate medium (McCoy 5A for HT-29 cells, RPMI 1640 for Caco-2 cells and PBS for the mucus assay) without supplements and adjusted to OD_600nm_ 0.25, which corresponded to approximately 10^8^ cells/mL. The bacterial suspensions were incubated for one hour on the epithelial cell monolayer or mucus at 37 °C in a CO_2_ incubator, followed by washing three times to remove the non-adherent cells. Adhered bacteria were lysed with 1% SDS-0.1 M NaOH and radioactivity was measured with a liquid scintillator (Wallac Winspectral 1414, Perkin Elmer, Waltham, MA, USA). The percentage of bound bacteria was calculated relative to the radioactivity of the bacterial suspension initially added to the wells.

### 2.5. Effect of Bacteria on Epithelial Integrity

Caco-2 cells undergo enterocytic differentiation and express intercellular junctional complexes, making them a suitable model to measure transepithelial electrical resistance (TER) as an indicator of the monolayer integrity [38,39,40]. The impact of bacterial isolates on the TER of Caco-2 monolayer was studied as previously described [29,37]. Briefly, 50,000 Caco-2 cells were seeded on PET inserts with a pore size of 3 um (Sarstedt, Nümbrecht, Germany) and grown for 8 days. The medium in the well was changed every 3 days. After 8 days Caco-2 cells form a monolayer, are partially differentiated and are in an appropriate growth phase to measure the development of transepithelial resistance and how it is affected by external factors [36,37]. *E. coli* TOP10 has been shown to adversely affect the monolayer integrity and was used as a negative control in the experiments [37]. Bacterial cells were washed once with RPMI medium and 100 μL of bacterial suspension adjusted to OD_600_ of 0.25 was added to the inserts after the time 0 measurement. The TER measurements were carried out using an EVOM epithelial voltohmmeter with an electrode (World Precision Instruments, UK). The inserts containing the bacterial isolates were incubated at 37 °C in a CO_2_ incubator and the TER was measured again after 24 hours. The blank resistance (measurement at time point 0) was subtracted from the measurements made after 24 hours of incubation, and the unit area resistance (Ω cm^2^) was calculated by multiplying the tissue resistance values by surface area of the filter membrane. The change in TER over 24 hours was calculated by comparing the samples to the medium control.

### 2.6. Genome Sequencing

Genomic DNA was extracted from bacterial cell pellets using the DNeasy Blood and Tissue Kit (Qiagen, Hilden, Germany) following the manufacturer’s protocol for Gram-negative bacteria. DNA was fragmented using a Q800R2 Sonicator (QSonica, Newtown, CT, USA) to approximately 500 bp, and genome libraries were prepared for paired end sequencing using the NEBNext^®^ Ultra™ II Kit (New England Biolabs, Ipswich, MA, USA) and quantified using the Library Quantification Kit (KAPA Biosystems, Waltham, MA, USA). Libraries were pooled in equimolar amounts and sequenced on the MiSeq (Illumina, Inc., San Diego, CA, USA). Read data were deposited in the NCBI SRA under BioProject PRJNA575760. Short read data were assembled using unmanned genome assembly pipeline (UGAP; https://github.com/jasonsahl/UGAP), which uses the SPAdes genome assembler. The species of each genome was identified using Kraken (PMID: 24580807) and confirmed by its placement in the phylogenetic trees generated for each species (see below).

Whole genome single nucleotide polymorphism (SNP) typing (WGST) was used to determine whether any genomes from this study had high identity to previously published genomes. For WGST, SNP matrices to identify point mutations among the isolates (and thus infer strain relatedness) were generated with NASP [41], in which reads were aligned to a publicly available assembly using BWA [42]. SNPs were called with Genome Analysis Toolkit (GATK) [43] and were only included in further analyses if they were a) present in all samples, b) covered by ≥10X depth with ≥90% consensus in each sample, and c) not in any duplicated regions in the reference genome as identified by NUCmer [44]. The resulting SNP matrix comprised the core genome common to all samples in the analysis. Maximum likelihood phylogenetic analyses were performed with IQ-TREE [45]. Maximum parsimony analyses were performed with MEGA v7.0 [46] or RAxML (version 8.2.10, [47]).

### 2.7. Statistical Analysis

All the adhesion and attenuation experiments were done using three to four technical replicates depending on the assay and repeated two to four times (biological replicates) to confirm the results. Different cultures of bacteria and different passages of cell lines were used in the separate experiments. A two-sample t-test was used to determine significant differences between a sample and the control. Homoscedasticity testing was performed with Levene’s test to identify equal or unequal variances. Point biserial correlation was calculated between the attenuation capacity as the dichotomous variable and adhesion or TER as the continuous variable. All statistical analyses were carried out with IBM SPSS Statistics program version 21.0 (IBM Corporation, New York, NY, USA) with a *p*-value of <0.05 considered statistically significant.

### 2.8. Data Availability

Whole genome sequence read data were deposited in the NCBI SRA under BioProject PRJNA575760.

## 3. Results

### 3.1. A High-Throughput Screening Method

In this study, we developed a high-throughput assay to screen commensal bacteria exerting anti-inflammatory action in the human gastrointestinal tract. First, bacteria from fecal material of a pre-screened, healthy FMT donor was cultivated on GAM (Gifu anaerobic medium), RCM (reinforced clostridial medium) and Brucella agar under anaerobic conditions and 600 bacterial isolates were picked during 72 hours of incubation (Figure 1). The screening of anti-inflammatory capacity was performed by pre-treating the HT-29 cell line first with the bacterial isolates and then with *E. coli* LPS, which induces an IL-8 release from the enterocyte cell line. In LPS-induced HT-29 cells that were pre-treated with an isolated bacterium as compared to controls without pre-treatment (i.e., LPS-induction alone), a decrease in the produced IL-8 levels was considered as an indication of anti-inflammatory capacity. In the first screening round, during which we tested 600 isolates with one replicate (well) per isolate, 170 isolates (28% of all the isolates) reduced the IL-8 levels as compared to the LPS control. The second screen using several replicates per isolate yielded 38 positive isolates that significantly reduced the IL-8 release from HT-29 cells. The positive isolates were purified and tentatively identified at the genus level by using 16S rRNA gene sequencing. Over 75% of the isolates belonged to the Bacteroidales order, including genera *Parabacteroides* (6 isolates), *Bacteroides* (22 isolates) and *Odoribacter* (1 isolate). The remaining isolates were identified as *Cutibacterium* (3 isolates), *Streptococcus* (1 isolate) and *Bifidobacterium* (5 isolates). We focused our further studies on the *Parabacteroides* and *Bacteroides* isolates, which formed a major part of the bacterial isolates and are specifically associated with the restoration of mucosal microbiota after FMT [20,21]. Moreover, these species have shown interesting immunomodulatory capacities in previous studies [12,19,48,49,50,51].

### 3.2. Whole Genome Sequencing (WGS)

The genomes of 28 *Parabacteroides* and *Bacteroides* isolates were obtained using whole genome sequencing (WGS) and identified on a species level with Kraken (PMID: 24580807; Appendix A). All six *Parabacteroides* isolates had a 5.0 Mb assembled genome size and 45.1% GC content and were confirmed as *Parabacteroides distasonis* (Pd). The *Bacteroides* isolates were identified as *B. caccae* (Bc, 3 isolates), *B. fragilis* (Bf), *B. intestinalis* (Bi, 3 isolates), *B. uniformis* (Bu), *B. vulgatus* (Bv, 5 isolates) and *B. ovatus* (Bo, 9 isolates) with assembled genome size and GC content ranging between 4.4–6.9 Mb and 41.9–46.5%. In addition, the parsimony trees based on whole genome single nucleotide polymorphism (SNP) typing (WGST) indicated a close relatedness of the isolates within a species (Figure 2 and Appendix A). Furthermore, SNP typing using NASP revealed that the isolates belonging to the same species had less than 10 genomic SNP differences to each other, but more than 10,000 SNP differences on average to database strains (Appendix A, Figure 2 and Appendix A). Thus, all isolates within the same species are most likely representatives of the same strain or at least have a common origin.

### 3.3. Anti-Inflammatory Properties

Next, we assessed the robustness of the immunomodulatory attributes among the 28 isolates, that is their ability to consistently alleviate LPS-induced IL-8 release from one experiment to another. The assay was repeated three or more times with all the purified and genome-sequenced bacterial isolates. Isolates that significantly (*p* ≤ 0.05) decreased IL-8 release in at least three individual experiments and in the majority of independent experiments (for example, in three out of five experiments) were considered to have attenuation competence (Table 1). All *B. caccae*, *B. intestinalis, B. uniformis* and *B. vulgatus* isolates repeatedly showed the ability to significantly decrease LPS-induced IL-8 release from HT-29 cells as compared to the LPS control. Among *B. caccae*, *B. intestinalis* and *B. vulgatus* isolates, the attenuation capacity seemed very stable and consistent across the isolates of the same species. The *B. fragilis* isolate could not be verified to exert anti-inflammatory action as it did not affect IL-8 production in LPS-induced HT-29 cells. Interestingly, only three out of six *P. distasonis* and two *B. ovatus* isolates were confirmed to exert anti-inflammatory action in vitro, suggesting that the attenuation competence is not a robust trait among these isolates, which are derived from a common ancestor strain and seem to be prone to loosing or inconstantly expressing the trait. Importantly, the non-attenuating isolates, where no significant (*p* ≥ 0.05) difference to the LPS control was noticed, did not stimulate a pro-inflammatory reaction in the HT-29 cell line either.

### 3.4. Mucosal Adherence of the Isolates

All *B. ovatus* isolates were clearly non-adherent to Caco-2 and HT-29 cell lines as well as to mucus since their relative adherence percentage (measured as the proportion of adhered bacteria compared to the total added bacteria) was below 1%, which can be considered unspecific binding comparable to background levels (Figure 3). The three *B. caccae* isolates adhered only to mucus. Among the *P. distasonis* strains, only Pd1 was adherent while the binding capacity of the other *P. distasonis* isolates was close to the background level. Conversely, all *B. intestinalis*, *B. uniformis* and *B. fragilis* isolates could adhere to Caco-2, HT-29 and mucus with a relative adherence level of approximately 2–5%. We found no correlation between the isolates’ capacities to adhere to HT-29 cells and to attenuate LPS-induced IL-8 release from the enterocyte cell line (Point biseral correlation, r_pb_ = 0.084, *p* = 0.670) and thus the anti-inflammatory action does not seem to require a firm cell-cell attachment between bacteria and intestinal epithelial cells.

Subsequently, we assessed the anti-inflammatory action of cell-free spent medium from the bacterial cultures by using the culture supernatant in the attenuation assay (Figure 4). Pre-incubation of HT-29 cells with 1:2 (50%) or 1:4 (25%) diluted spent culture medium from the isolates significantly decreased IL-8 release during the subsequent LPS stimulation as compared to the LPS and unspent bacterial growth medium controls, thus confirming that a firm cell-cell attachment between bacteria and epithelial cells is not necessary to exert the anti-inflammatory effect.

### 3.5. Epithelium Reinforcing Action

Next, we studied the epithelium reinforcing properties of the isolates that showed anti-inflammatory properties. The ability of the isolates to enhance Caco-2 monolayer integrity was assessed by co-culturing the enterocyte cell line with the bacterial isolates and using transepithelial electrical resistance measurements to address maintenance or improvement of the barrier function (TER, Figure 5). A commercial *E. coli* laboratory strain was used as a negative control as it adversely affects the integrity of the monolayer [37]. After eight days of post-plating the Caco-2 monolayer showed a baseline TER of 319 ± 16 Ω. Most isolates that displayed the capacity to attenuate LPS-induced IL-8 release also showed the capacity to reinforce epithelial integrity in vitro (Figure 5). As an exception, the three *P. distasonis* isolates did not induce significant changes in TER values as compared to the monolayer treated with only medium. In other words, the isolates did not reinforce nor compromise the epithelial integrity of the monolayer. *B. vulgatus* isolates improved the monolayer integrity the most effectively. Among the isolates, the capacity to attenuate LPS-induced IL-8 production correlated positively with the ability to reinforce the integrity of Caco-2 monolayer (r_pb_ = 0.397, *p* = 0.061). However, there was no correlation between the adherence level of the isolates to Caco-2 cells and their ability to reinforce the enterocyte monolayer integrity (r = 0.109, *p* = 0.620).

### 3.6. Genes Involved in Host-Microbe Interactions of Bacteroides spp.

We screened the assembled genomic data of the isolates for genes known to have an important role in the host-microbe interactions of *Bacteroides* spp. First, we searched the genomes of the isolates for genes that encode enzymes for Kdo2-lipid A modification using BLAST (Basic Local Alignment Search Tool, blastp algorithm). All the genomes of *Parabacteroides* and *Bacteroides* isolates carried the genes encoding LpxA-LpxK for the Kdo_2_-lipid A moiety synthesis but lacked the genes for LpxL and LpxM acyltransferases. LpxL and LpxM are needed for adding the secondary acyl chains (hexa-acylation) into lipid A yielding a highly pro-inflammatory LPS structure typical for gamma-Proteobacteria including *E. coli* [52,53]. Inability to hexa-acylate lipid A potentially explains why the *Parabacteroides* and *Bacteroides* isolates, despite being LPS-carrying organisms, did not evoke pro-inflammatory responses in enterocytes.

As we found the attenuation capacity to be independent of the cell-cell contact and possibly mediated through secreted effector molecules such as zwitterionic capsular polysaccharides (ZPS, e.g., Polysaccharide A of *B. fragilis*) transported in OMVs, we searched the genomes for *wcfR* gene homologues encoding for a protein that is needed in the synthesis of ATTGal-ZPS proteins. A match to the *wcfR* gene was found in the genome of the *B. fragilis* isolate (90.9% nucleotide identity, data not shown) but not in the other isolates and thus could not explain their anti-inflammatory properties.

We also searched the genomes of the *Parabacteroides* and *Bacteroides* isolates for a gene encoding serine palmitoyltransferase (SPT, BT_0870), an enzyme mediating the synthesis of sphingolipids, which were recently reported to mediate the *Bacteroides*-host interaction in maintaining homeostasis and symbiosis in the gut [24]. A match of 87%, 88% and 76% nucleotide identity to the *spt* gene was found in the genomes of all *B. caccae*, *B. ovatus* and *B. intestinalis* isolates, respectively. A 73% match to *spt* was also found in the genome of *B. fragilis*. No significant similarity was observed in *P. distasonis*, *B. uniformis* or *B. vulgatus* genomes concerning the *spt* gene. Thus, the presence or absence of the *spt* gene did not explain the anti-inflammatory or epithelium re-enforcing capacities of the isolates. Overall, we could not confirm across all our isolates the presence of *wcfR* or *spt*, which have been previously linked with the anti-inflammatory properties of *Bacteroides* spp. Thus, these genes do not seem to be universally distributed among *Bacteroides* spp. and the properties encoded by them can be suspected to mediate anti-inflammatory properties in some but not all strains.

## 4. Discussion

In this study we developed a high-throughput method to screen gut commensals with anti-inflammatory potential and retrieved in pure cultures 38 isolates, the majority of which were representatives of the genera *Bacteroides* and *Parabacteroides*. We acknowledge that the screening in our study was limited to those bacteria which can be grown under the culture conditions applied, but there are multiple species present in the human gut microbiota that can exert anti-inflammatory action, including also potentially novel species [8]. *Bacteroides* and closely related genera are highly abundant commensals in the human gut and their decreased abundance has been linked with intestinal inflammation e.g., in IBD and cystic fibrosis patients [11,12,54,55], and their higher abundance, specifically *B. fragilis and B. finegoldii*, was linked to a donor stool batch that was effective in keeping ulcerative colitis (UC) in remission after FMT [56]. Furthermore, several mice studies have recently demonstrated the potential of *Bacteroides* spp. in ameliorating intestinal inflammation [19,48,51,57,58]. Thus, the potential of *Bacteroides* spp. and closely related bacteria to alleviate inflammation in vitro seems to correspond to their in vivo action.

Based on the WGS results, our screened isolates from an FMT donor were identified as *P. distasonis*, *B. caccae*, *B. fragilis*, *B. intestinalis*, *B. uniformis*, *B. vulgatus* and *B. ovatus,* which are all typical gut commensals in the human intestinal tract. SNP typing revealed that isolates belonging to the same species are representatives of the same strain or at least have a common ancestor. Our results and conclusions are in line with the previous observations that single strains typically dominate most species in the gut microbiota [59]. Concerning *Bacteroides* spp., even in the case of multiple strains within a species co-existing, the relative abundance of the main strain was 87% and 91% for *B. uniformis* and *B. vulgatus*, respectively [59]. Notably, *Bacteroides* and *Parabacteroides* species are remarkably genetically consistent with only on average 0.45% of nucleotide variants between strains, which may hamper strain differentiation. On the other hand, with more genetically plastic gut commensals such as *Prevotella*, the variation reaches to 2.44% [59]. However, our isolates had less than 10 SNP differences with each other within the same species and thus we considered them to have a common ancestor or represent the same strain.

The in vitro capacity to attenuate inflammation was shown to be consistent among the isolated strains within the species *B. caccae*, *B. instestinalis* and *B. vulgatus* as all the isolates significantly decreased the cytokine IL-8 levels in an LPS-induced enterocyte cell line. In the case of *P. distasonis* and *B. ovatus*, the attenuation competence varied among the isolates despite them being genomically very similar, and the reason for this remains to be studied in detail. Despite consistency in the cultivation conditions, we cannot rule out variations in gene expression in different cultivation batches. Another possibility is that some of the isolates have lost their ability to produce the needed effector molecules, which remains to be studied with more detailed comparative genomic approaches.

Three out of the six *P. distasonis* isolates were able to attenuate inflammation in our in vitro model, although they did not enhance the epithelial integrity in the Caco-2 monolayer. In general, the anti-inflammatory attributes among gut commensals are highly strain-specific, but *P. distasonis* strains with anti-inflammatory properties have been described previously. A recent study showed that the membrane fraction of *P. distasonis* caused a 55% and 29% reduction in IL-8 production in *E. coli* LPS induced HT-29 and SW480 cell lines, respectively [48]. *P. distasonis* crude lysate and membrane fraction have also been shown to decrease the disease severity in a dextran sulfate sodium (DSS)-induced colitis murine model [60].

We did not find any correlation between the isolates’ ability to adhere to intestinal epithelium and to attenuate LPS-induced inflammation or to strengthen the epithelial monolayer integrity. Yet, we observed that a firm cell-cell attachment between bacteria and enterocytes was not necessary for the isolated strains to exert their anti-inflammatory activity as their culture supernatants also attenuated LPS-induced IL-8 release from the HT-29 cell line. We did not address the bacterial cell viability during the incubation time with the HT-29 cells, but as bacterial supernatants were also found to exert the anti-inflammatory action, it seems that the presence of effector molecules rather than bacterial viability per se is needed for the activity. Secretion of effector molecules, such as polysaccharide A, and their transportation, for example in OMVs, to enterocytes provides an explanation for host-microbe interaction in distance without direct physical interaction [61]. In the intestine, bacterial OMVs can also travel through the thick mucus layer and even through the intestinal epithelial layer. As an example, OMVs secreted by *B. thetaiotaomicron* have been found localized in the host immune cells in the intestinal mucosa [62]. Moreover, *B. vulgatus* has been shown to modulate immune cells by producing OMVs, which diffuse through the mucin layer and induce tolerance in dendritic cells in a TLR4 and TLR2 dependent manner [49]. Besides bacterial effector molecules, short-chain fatty acids (SCFAs) produced by bacteria can regulate the release of LPS-induced IL-8 from epithelial cells [63].

The identification of bacterial effector molecules responsible for exerting anti-inflammatory action in the gut still remains an open question. We searched the genomes of our *Bacteroides* and *Parabacteroides* isolates for *wcfR* gene, which is needed for the synthesis of ZPS (e.g., PSA), but only identified the homologue in the genome of the *B. fragilis* isolate. Interestingly, we discovered *B. thetaiotaomicron spt* gene (BT_0870) homologues with a significant similarity in the genomes of *B. caccae*, *B. intestinalis*, *B. fragilis* and *B. ovatus*. The *spt* gene is involved in the synthesis of sphingolipids, which mediate bacteria-host interactions with the help of bacterial OMVs [64]. *Bacteroides* species are known sphingolipid producers in the human intestinal tract but to what extent they contribute to the gut homeostasis still remains to be solved. Colonization of germ-free mice with sphingolipid-deficient *B. thetaiotaomicron* led to intestinal inflammation and altered ceramide pools, suggesting that bacterial sphingolipids have a major effect on intestinal homeostasis [24].

*Bacteroides* and *Parabacteroides* spp. are Gram-negative bacteria containing LPS, which is known as a potent endotoxin that induces a strong pro-inflammatory reaction in the host. However, *Bacteroides* LPS has profoundly different properties than the more toxic LPS of Enterobacteriaceae, including *E. coli* [65]. The lipid A moiety in the LPS structure is responsible for its degree of endotoxicity affecting the ligand affinity to TLR4-MD2 complex and downstream activation of the NF-kB pathway [66]. *Bacteroides* species seem to possess under-acylated, less toxic lipid A structures based on mass spectrometry analysis which showed the presence of penta- and tetra-acylated lipid A forms, whereas *E. coli* possesses the highly pro-inflammatory, hexa-acylated lipid A domain [65]. Our WGS analysis showed that *Bacteroides* and *Parabacteroides* isolates of this study lack the genes for LpxL and LpxM needed to construct a hexa-acylated lipid A. On the contrary to the pro-inflammatory form of *E. coli* LPS, the LPS of *Bacteroides* species is immunosuppressive [65,67]. Indeed, *Bacteroides* LPS has been shown to modulate the response of primary human peripheral blood mononuclear cells (PBMCs) to *E. coli* LPS stimulation by reducing the production of proinflammatory cytokines, such as TNF-a and IL-6 [65]. Moreover, weak agonistic *Bacteroides* LPS was shown to have inflammation-reducing properties by ameliorating inflammatory immune responses in an experimental colitis mouse model [68]. Thus, *Bacteroides* LPS may be in part responsible for the anti-inflammatory properties observed in this study.

Bacteroidales spp., such as *Alistipes onderdonkii*, *B. fragilis* and *B. thetaiotamicron*, were shown in monocolonised mice to maintain intestinal homeostasis by promoting intraepithelial lymphocytes (IEL) in the colon [51]. IL-6 secretion by IELs is involved in promoting barrier function and requires bacterial MyD88 signaling. The Bacteroidales species increased IL-6 production in IELs in vivo compared to an *E. coli* control, yet only with *B. thetaiotamicron* was the increase significant. Furthermore, *B. fragilis* and *B. ovatus* were shown to relieve LPS-induced inflammation in a mouse model by decreasing TNF-α and increasing IL-10 cytokines as well as restoring the Treg/Th17 balance [58]. Promising results concerning the effectiveness of bacteriotherapy in treating IBD were acquired in a recent study by Ihekweazu et al. (2019) where *B. ovatus* monotherapy ameliorated colitis in a murine model and enhanced epithelial recovery more effectively than traditional FMT [19]. The potential of novel isolates to exert anti-inflammatory action in vivo remains to be addressed, but the results from previous studies seem encouraging in this respect. The commensal *Bacteroides* and *Parabacteroides* isolates with anti-inflammatory and epithelium enhancing capacity of this study are potential candidates concerning bacteriotherapeutic applications aiming to restore the gut homeostasis. The epithelium reinforcing action of the isolates was shown to enhance Caco-2 monolayer integrity as compared to medium. However, the isolates capacity to protect against detrimental effects of external factors on the epithelial integrity or to restore the integrity after an insult was not studied and could be addressed in future studies.

In conclusion, we developed a high-throughput screening method to isolate bacterial strains exerting anti-inflammatory capacity in vitro. In this study we isolated from a healthy fecal donor *Bacteroides* and *Parabacteroides* species with capacity to alleviate *E. coli* LPS-induced IL-8 production from enterocytes and enhance the epithelial monolayer integrity. Neither of these attributes correlated with the adhesion ability of the isolates, yet the attenuation capacity sustained when spent medium from isolates was used suggests that the anti-inflammatory competence is independent of a cell-cell contact between the bacterium and epithelium. In future studies, we aim to identify potential effector molecules involved and study the role of OMVs in their anti-inflammatory capacity, for which the acquired WGS data provide an ideal starting point. The potential of bacteriotherapy for the treatment of different conditions is becoming more evident, thus underlining the importance of identifying anti-inflammatory commensals, such as *Bacteroides* spp., and their effector molecules.

## Figures and Tables

**Figure 1 nutrients-12-00935-f001:**
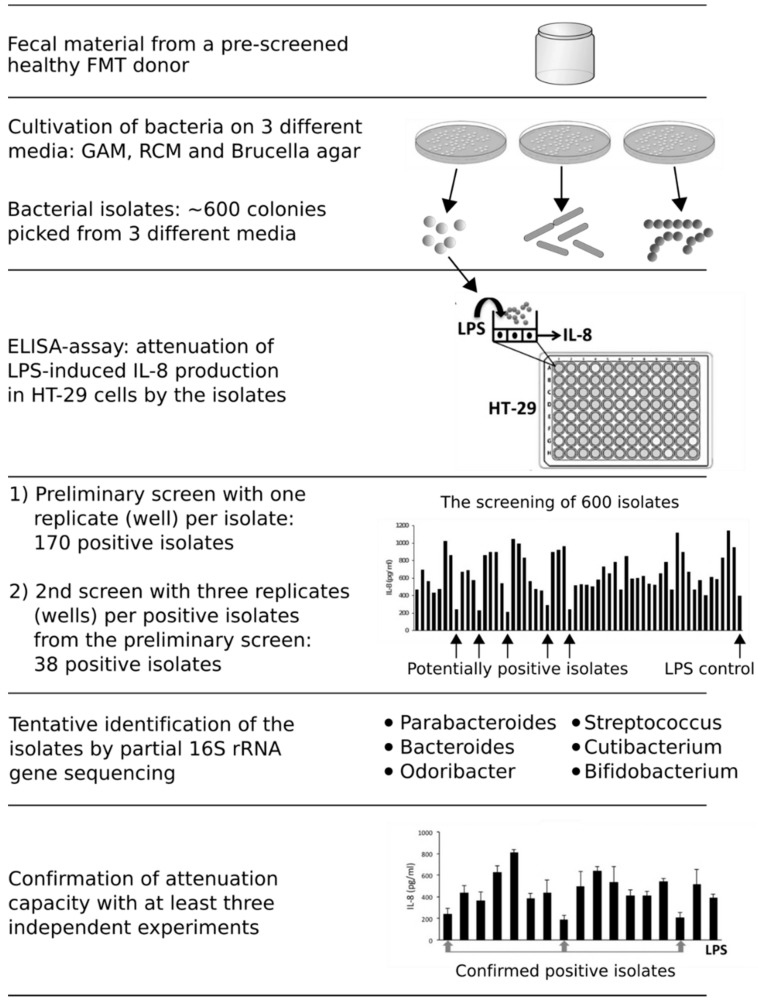
The high-throughput approach to screen for anti-inflammatory bacteria and the outcomes from different steps.

**Figure 2 nutrients-12-00935-f002:**
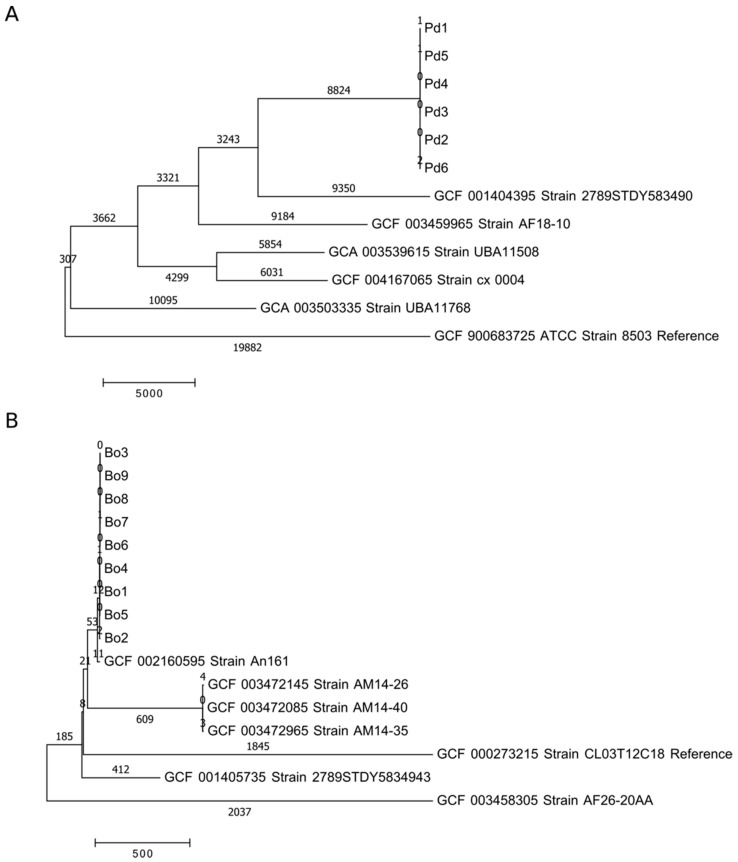
Estimates of evolutionary divergence by whole genome single nucleotide polymorphism (SNP) typing (WGST) (number of SNP differences per sequence) between the isolates belonging to the same species and previously published genomes (NCBI database). Panel (**A**) is *P. distasonis* (Pd) and panel (**B**) is *B. ovatus* (Bo).

**Figure 3 nutrients-12-00935-f003:**
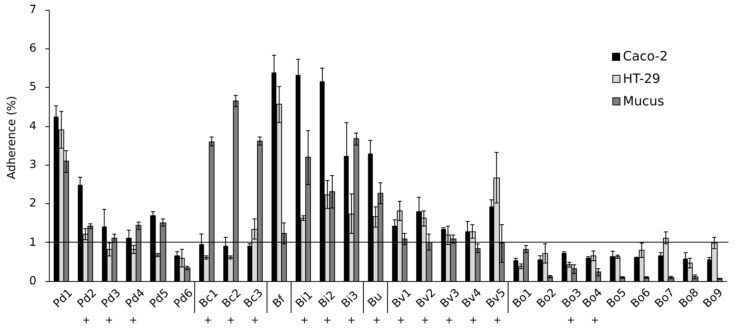
Adherence of the isolates to Caco-2 and HT-29 cell lines and to intestinal mucus. Pd = *P. distasonis*, Bc = *B. caccae*, Bf = *B. fragilis*, Bi = *B. intestinalis*, Bu = *B. uniformis*, Bv = *B. vulgatus*, Bo = *B. ovatus*. Numbers 1–9 refer to the specific isolate. Symbol + indicates isolate’s capacity to attenuate LPS-induced IL-8 release from HT-29 cell line.

**Figure 4 nutrients-12-00935-f004:**
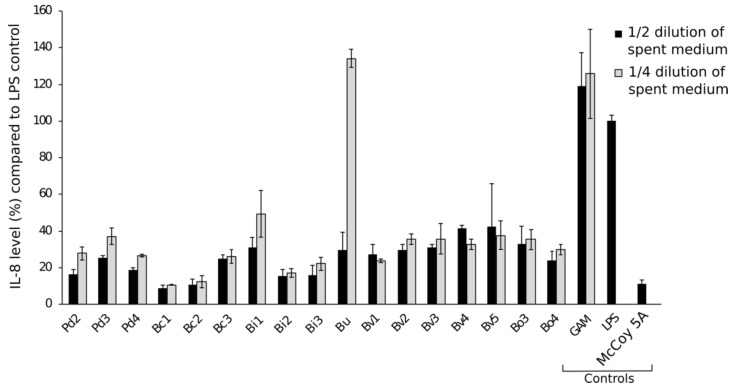
The attenuation capacity of the isolates using spent medium i.e., culture supernatant. Gifu anaerobic medium (GAM) and McCoy 5A media as well as only LPS were used as controls. LPS control represents 100% IL-8 release from HT-29 cells. ½ and ¼ dilutions of spent media were used in the assay. Pd = *P. distasonis*, Bc = *B. caccae*, Bf = *B. fragilis*, Bi = *B. intestinalis*, Bu = *B. uniformis*, Bv = *B. vulgatus*, Bo = *B. ovatus*. Numbers 1–9 refers to certain isolates.

**Figure 5 nutrients-12-00935-f005:**
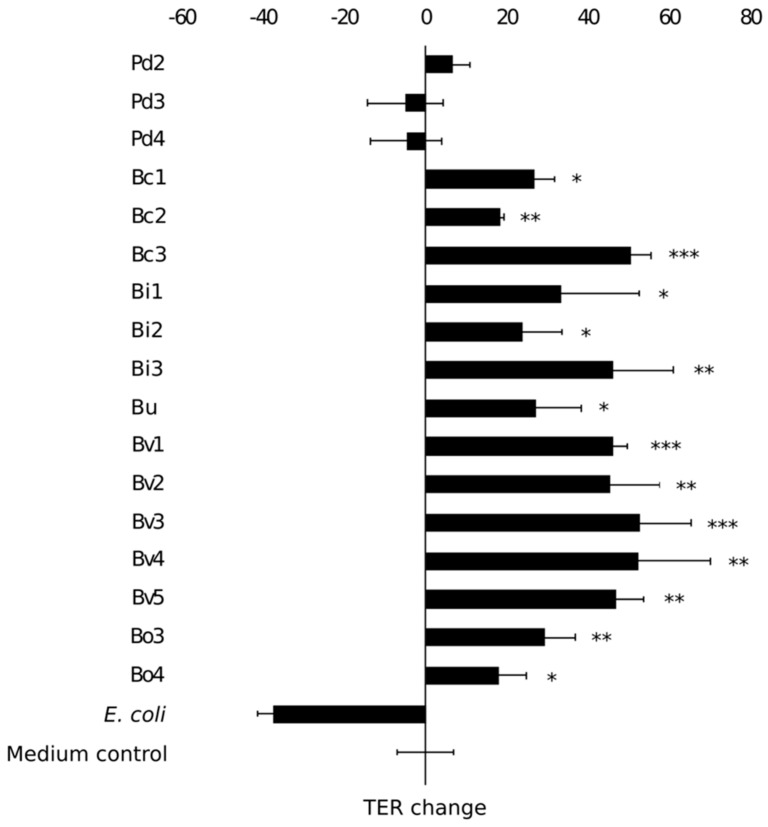
The effect of isolates on transepithelial electrical resistance (TER) of Caco-2 monolayer. *E. coli* was used as a negative control. Results are shown as the mean of TER value change (Ω/cm^2^) in 24 hours and standard deviation of three technical replicates (parallel wells). An asterisk indicates significantly different (*p* < 0.05) value as compared to the untreated Caco-2 monolayers (medium control). * *p* < 0.05, ** *p* < 0.01, *** *p* < 0.001. Pd = *P. distasonis*, Bc = *B. caccae*, Bi = *B. intestinalis*, Bu = *B. uniformis*, Bv = *B. vulgatus*, Bo = *B. ovatus*. Numbers 1–9 refer to the specific isolate.

**Table 1 nutrients-12-00935-t001:** The capacity of *Bacteroides* and *Parabacteroides* isolates to attenuate lipopolysaccharide (LPS)-induced IL-8 release from HT-29 cell line.

Isolate	Significant IL-8 Decrease */ Total Number of Attenuation Experiments	% Decrease in IL-8 Release as Compared to the LPS Control	Attenuation Competence ^#^
Pd1	1/4	7–53	-
Pd2	4/5	35–61	+
Pd3	3/3	27–39	+
Pd4	4/6	23–59	+
Pd5	0/3	0–11	-
Pd6	0/3	15–23	-

Bc1	4/6	12–44	+
Bc2	3/4	21–46	+
Bc3	3/5	33–48	+

Bf	0/3	1–34	-

Bi1	3/3	21–72	+
Bi2	3/4	25–66	+
Bi3	3/4	27–51	+

Bu	4/6	34–59	+

Bv1	3/4	26–39	+
Bv2	3/3	23–36	+
Bv3	5/9	39–59	+
Bv4	3/3	25–40	+
Bv5	3/3	41–55	+

Bo1	2/5	5–58	-
Bo2	4/8	0–39	-
Bo3	4/6	30–50	+
Bo4	3/4	35–57	+
Bo5	3/7	0–39	-
Bo6	0/3	17–26	-
Bo7	3/7	0–28	-
Bo8	2/5	0–54	-
Bo9	1/5	0–45	-

* Significant (*p* < 0.05) decrease in IL-8 release as compared to the control, i.e., LPS stimulation without a prior exposure to the studied isolate. #no attenuation or seldomly detected (−), attenuation constantly detected (+). Pd = *P. distasonis*, Bc = *B. caccae*, Bf = *B. fragilis*, Bi = *B. intestinalis*, Bu = *B. uniformis*, Bv = *B. vulgatus*, Bo = *B. ovatus*. Numbers 1–9 refer to the specific isolate. # Isolates that significantly (*p* ≤ 0.05) decreased IL-8 release in at least three independent experiments and in majority of experiments were considered to have attenuation competence (+).

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
