# Peer review of "Isolation of Anti-Inflammatory and Epithelium Reinforcing Bacteroides and Parabacteroides Spp. from A Healthy Fecal Donor"

_nutrients, 2020, doi:10.3390/nu12040935_

Round 1
Reviewer 1 Report
Comments for the manuscript (utrients-745925) entitled “Isolation of anti-inflammatory and epithelium reinforcing Bacteroides and Parabacteroides spp. from a healthy fecal donor”.
This manuscript describes the development of an in vitro screening assay to isolate intestinal commensal bacteria from healthy fecal donor. Several potential applications could be developed from this kind of design. However, they focus their study mainly in Bacteroides, opportunistic bacteria increased in pathologies, such as metabolic diseases. Despite Bacteroides are part of a healthy microbiota, in humans, high fat/protein diets have been connected with higher levels of Bacteroides (compare to Prevotella, increased in high-fiber diets). In my opinion, to define Bacteroides as ‘promising bacteriotherapeutic candidates’ (Line 73) is overstatement.
Another major concern is the missing of cell viability assay to corroborate that the changes in IL-8 concentration are only due to the isolates. Please, include this new assay in the next version of the paper.
Following minor points also concerned:
Line 67. Please, include more information about Bacteroides, including an state-of-art about this genus.
Line 84. Please, replace the sentence ‘Feces from healthy […] for FMT’ before the sentence ‘The frozen fecal solution was thawed…’
Line 88. Human microbiota depends on age, body weight, antibiotics consumption, diet… Please, include these donor characteristics.
Line 98. ‘… as described below and in the paragraph 2.3 using one replicate.’ Is confusing. Please, concrete it.
Line 118. Gibco’s country is missing. Please, complete.
Line 201. Parametrics test as t-test can be applied only in normal distributions. Please, indicate if this has been analyzed.
Line 269. Improve the quality of the table, following similar format to the rest of the text
Reviewer 2 Report
Dear Authors
This manuscript about the isolation of Bacteroides and Parabacteroides with anti-inflammatory and epithelium reinforcing activity is clear and well written. I just have some minor observations:
- Figure 1 looks like half-table or half-figure, consider the design of an algorithm instead.
- Page 11, line 365, the citation format is incorrect.
- Describe why did you choose HT-29 and Caco 2 cells instead of other in vitro models.
Reviewer 3 Report
OVERALL
In the present study, the authors screened fecal isolates from a healthy donor for anti-inflammatory activities in colon epithelial cell models. Isolates with activity were further screened and their activities probed. Promising isolates which decreased pro-inflammatory cytokine levels and enhanced barrier integrity were identified. Overall, the results are novel, timely and relevant. However, methodological concerns (particularly the statistical analyses) need to be addressed.
INTRODUCTION
Line 39
The inclusion of “processed foods” here is too generic. There are no foods that are not processed unless you are going to your garden and eating it directly out of the ground (washing a carrot is “processing”). What does that term mean? Seems irrelevant to the manuscript
METHODS
Line 84-86
How long was the sample exposed to air following collection? Please provide a brief paragraph regarding collection and storage and how an anaerobic environment was maintained
Line 87
Please provide some basic information about the donor here (age, sex, BMI, antibiotic or other medication and supplements use, basic health history, etc.)
The use of a single donor could be a weakness. Please address this
Line 92-93
Please clarify what you define as “anaerobic conditions” (gas mixture, elimination of residual oxygen, approximate oxygen levels in the chamber, sparging of solutions, etc.)
Line 96
Please clarify what is meant by “appropriate broth or on agar” and how such determinations were made
Line 99
Why was n=1 used for initial screening of anti-inflammatory isolates?
Line 129
How was this concentration of bacteria selected?
Please comment on the relevance of these bacterial concentrations to levels that these individual isolates can reach in the colon lumen or mucosa.
Does “8 days old” mean 8 days after seeding?
Line 130
Clarify the rationale for using aerobic conditions…you are primarily looking at the interaction of dead and/or dying bacterial cells with the colon cells
Line 155, 274
Please clarify…porcine mucus was added to plates that already had cells growing in them? Based on line 274 it sounds like separate experiments with epithelial cells and mucus were performed, but this is not clear in the methods
Line 169
8 days is probably not long enough to fully differentiate Caco-2 cells. 14-21 days is much more common. This is a flaw that needs to be addressed.
Line 171-175
Please describe TEER measurements
Line 201-202
Due the large number of isolates screened, the two-sample t-test is not conservative enough to provide reasonable protection against false positives. At minimum, Dunnett’s test should be used to compare multiple treatments to controls while factoring in multiple comparisons. Bonferroni would be appropriate here as well.
RESULTS
Line 212, 348
I think a major caveat is that this assay screened only bacteria cultivable outside the colon.
Line 221-223
I think it is admirable that 600 isolates were tested, and I can see why n=1 was used due to practical concerns. However, what was the criteria for determining that a specific isolate reduced IL-8 secretion? Was a minimum % reduction used to minimize the number of “hits”?
Please make the initial screening data available in supplementary files
If 170 isolates were compared to control, a test to control for false positives absolutely has to be employed. The two-sample t-test is not conservative enough to provide reasonable protection against false positive (it is the least conservative test). The statistical analysis needs to be redone with correction for multiple comparisons. It is ok to report what you have here…but identify how many of the 38 identified isolates would pass a FDR correction? Show the analysis when accounting for multiple comparisons.
The results of the 170 isolates used for secondary screening need to be shown in tabular or heatmap form…we need to see the data. I know it is a lot of data, but I think a heatmap could work.
Line 257-258
Again, I think for 28 isolates, the risk of false positives is high enough that precautions need to be taken against false positives. Please perform a secondary analysis using an appropriate post hoc test or FDR and report the results.
Line 283, 329, 390
Contact is not the same as adherence, so I would soften these statements significantly
Line 285-290
Please show these data
Line 302-304, Figure 4
Again, need to account for multiple comparisons.
One criticism of the epithelial integrity assay is that no challenge (such as DSS) was used to degrade the barrier and observe the ability of the isolates to preserve integrity. An already healthy and functional barrier does not need to be improved, just maintained. So this needs to be addressed as a limitation.
Furthermore, we need to know what the baseline TEER value was for each treatment in order to understand what the state of the barrier was prior to treatment.
DISCUSSION
Line 380-381
See my comment above…barrier integrity was not challenged, so the meaning of this assay is not as clear as it would have been with a challenge. Please address this.
TABLES
Table 1
Please list % IL-8 decrease ranges for all isolates….including those that had no effect or even increased IL-8 secretion
Specify the criteria used for “attenuation competence” designations
Round 2
Reviewer 1 Report
Thank you for your answers.
Author Response
We would also like to thank the reviewer for the efforts considering our manuscript.
Reviewer 3 Report
The authors partially address the anaerobic conditions. The authors still need to include chamber purging times and whether residual O2 was measured.
The authors provided valuable insight regarding the concentration of the bacteria used, in the response, but this information needs to be included in the actual manuscript so the readers understand the rationale.
The authors need to include more of the justification of aerobic conditions for the HT-29 cell assay that was provided in the response, in the actual manuscript text. This information needs to be included in the actual manuscript so the readers understand the rationale.
Author Response
Dear Editor and Reviewer,
We would like to thank you for your efforts concerning our manuscript. We have addressed the minor revisions.
Point 1. The authors partially address the anaerobic conditions. The authors still need to include chamber purging times and whether residual O2 was measured.
Response 1. The Ruskinn anaerobic chamber automatically purged gas inside the chamber in order to retain anaerobic conditions. If any O2 remained or entered to the anaerobic chamber during its use, it was scavenged by a palladium catalyst situated inside the chamber under the floor tray. The O2 reacts with the H2 forming water. Also anaerobic color indicator strips were used inside the chamber to verify anaerobic conditions as recommended by the manufacturer.
We have added this information to the materials and methods.
Lines 103-106 “The anaerobic chamber automatically purged gas inside the chamber when needed. Palladium catalyst inside the chamber scavenged any residual O2 and anaerobic color indicator strips were used to verify anaerobic conditions.”
Point 2. The authors provided valuable insight regarding the concentration of the bacteria used, in the response, but this information needs to be included in the actual manuscript so the readers understand the rationale.
Response 2. We thank the reviewer and added this information to the manuscript.
Lines 144-149: “The bacterial concentration used in the experiments (final concentration of 107 cells/ml) was considered biologically relevant for in vitro assays as the amount of Bacteroides spp. in the intestine could theoretically be 5 x 109/ml up to 3 x 1010/ml: the abundance of Bacteroidetes in a healthy microbiota has a lot of inter-individual variance, but in general it is up to 30% in feces and the bacterial content in the stool is approximately 1011/g of feces [30]. The percentage of Bacteroides spp. in the feces of the donor used in this study was approximately 5% [21].”
Point 3. The authors need to include more of the justification of aerobic conditions for the HT-29 cell assay that was provided in the response, in the actual manuscript text. This information needs to be included in the actual manuscript so the readers understand the rationale.
Response 3. We appreciate the reviewer’s suggestions and added more information to the manuscript:
Lines 153-158: “The use of aerobic conditions with increased CO2 (5%) in the screening assay using anaerobic bacteria was considered appropriate due to the short activation time i.e. 1 hour incubation on the HT-29 cells before removing the bacterial cells and adding LPS. Also, it is known that anaerobic bacteria can tolerate oxygen for 1 hour up to 72 hours depending on the species [31] and Bacteroides spp. sustain aerobic conditions better compared to other anaerobic gut bacteria [32].”